# Imaging and Clinical Outcomes Six Months After Middle Meningeal Artery Embolization with Squid for Chronic Subdural Hematoma: A Prospective Study

**DOI:** 10.3390/diagnostics15111424

**Published:** 2025-06-03

**Authors:** Ángela H. Schmolling, Carlos Pérez-García, Isabel Bérmudez, Alfonso López-Frías, Eduardo Fandiño, Carmen Trejo, Santiago Rosati, Daniel Padrón, Lara Guardado, José Carlos Méndez, Juan Arrazola, Manuel Moreu

**Affiliations:** 1Departamento de Neurorradiología Intervencionista, Servicio de Radiodiagnóstico, Hospital Clínico San Carlos, Calle Profesor Martín Lagos s/n, 28040 Madrid, Spain; anghuete@gmail.com (Á.H.S.); manumoreu@gmail.com (M.M.); 2Departamento de Neurorradiología Intervencionista, Servicio de Radiodiagnóstico, Hospital Clínico Ramón y Cajal, M-607, km. 9, 100, Fuencarral-El Pardo, 28034 Madrid, Spain; 3Servicio de Geriatría, Hospital Clínico San Carlos, Calle Profesor Martín Lagos s/n, 28040 Madrid, Spain

**Keywords:** middle meningeal artery, chronic subdural hematoma, EVOH, Squid

## Abstract

**Background:** Chronic subdural hematoma (CSDH) is a common condition in older adults with rising rates of incidence. While burr hole drainage remains the standard treatment, it is associated with significant recurrence and complications. This study assesses MMA embolization with Squid, both as a standalone procedure and as an adjunct to surgery. **Methods:** Our prospective registry included 101 patients with 134 CSDH cases treated at two tertiary care centers from December 2020 to January 2024. Patients were divided into two groups: embolization alone and embolization combined with surgery. Demographic, clinical, radiological, and procedural data were collected. Follow-up imaging was conducted at 1, 3, and 6 months. Treatment failure was defined as rescue surgery, hematoma thickness ≥ 10 mm, midline shift > 3 mm at 6 months, or procedure-related death. **Results:** Fifty-two patients (51.5%) underwent combined treatment, and forty-nine (48.5%) received embolization alone. Most were men (68.3%) and the median age was 82 years. Combined-treatment patients had larger hematomas and more symptoms. Procedures were performed under general anesthesia in 72.3% of patients, with radial and femoral access used equally frequently, and 32.7% underwent bilateral embolization. Patients’ hematoma thickness in follow-up imaging showed a significant decrease (*p* = 0.000), reaching a median of 0 mm at six months, with no significant difference between groups. Complications occurred in 5.9%, and treatment failure in 4%. Mortality was higher in the embolization-only group, likely reflecting greater rates of comorbidities. **Conclusions:** This study supports the use of MMA embolization with Squid as a safe and effective treatment for CSDH. Comparable procedural and radiological outcomes in both groups suggest embolization alone may suffice in select patients, offering a less invasive alternative.

## 1. Introduction

Chronic subdural hematoma (CSDH) is an increasingly common neurological condition that particularly affects the elderly, with incidence rates that have risen significantly over the past three decades [1]. Traditionally, the standard treatment for CSDH has been surgical evacuation, which is typically performed via burr hole drainage [2]. Although this approach effectively alleviates the mass effect exerted by the hematoma, it is associated with notable limitations, including an recurrence rate estimated to be around 11% in the literature [3] and potential complications, especially in older adults or those with comorbidities. These challenges highlight the need for alternative therapeutic strategies that address the limitations of surgery, reduce recurrence, and improve patient outcomes.

Recent advances in understanding the pathophysiology of CSDH have led to the emergence of novel treatment approaches within the field of interventional neuroradiology. Traditionally, CSDH was attributed to traumatic brain injury leading to the rupture of bridging veins. However, recent studies propose a new hypothesis that implicates chronic inflammation as a central mechanism in CSDH development [4,5]. Unlike the traditional model, this hypothesis suggests that CSDH originates from an injury to the capillary-rich dural border cell layer—a soft tissue layer at the dura-arachnoid interface—causing persistent inflammation, membrane formation, and the leakage of blood and fluid into the subdural space [4,5,6]. This chronic inflammatory process is believed to contribute to the formation of membranes with ongoing angiogenesis, resulting in a well-vascularized cavity that gradually fills with fluid and blood, ultimately leading to clinical symptoms weeks after the initial injury [1].

A promising therapeutic innovation stemming from this understanding is middle meningeal artery (MMA) embolization. This minimally invasive procedure is designed to interrupt the inflammatory cascade within the dural border cell layer, controlling capillary bleeding and facilitating the resolution of the hematoma. By addressing the underlying inflammatory process, MMA embolization can significantly reduce recurrence rates, particularly in patients with risk factors such as anticoagulant or antiplatelet therapy [2,7,8,9,10]. It offers a viable alternative for patients who are poor candidates for surgical intervention and obviates the need to interrupt prior treatments [11]. This makes MMA embolization a versatile treatment option that can serve as a primary therapy, as an adjunct to surgery, or as a rescue intervention for recurrent hematomas.

The past few years have seen a surge in research exploring the effectiveness and safety of MMA embolization, both as a standalone treatment and in combination with surgical intervention. Despite promising initial findings, further data are needed to clarify the role of this technique in routine clinical practice, particularly in terms of patient selection, optimal procedural techniques, and long-term outcomes. This study aims to comprehensively review all cases of MMA embolization used for CSDHs treated at two tertiary care centers, assessing its effectiveness as both a primary and adjunct treatment strategy and providing insights into its potential role in the evolving landscape of CSDH management.

## 2. Materials and Methods

A prospective registry was created of all cases of MMA embolization performed at two tertiary hospitals between December 2020 and January 2024. The study included 101 patients and a total of 134 hematomas. A dual comparative analysis was performed, dividing each cohort into two groups: those treated with embolization alone and those who received embolization combined with surgical intervention. Demographic, clinical, radiological, and procedural data were collected, with follow-up imaging and clinical evaluations conducted at 1, 3, and 6 months post-treatment.

The study population comprised patients who had undergone MMA embolization as either a standalone treatment, concurrent with surgical evacuation during the same hospital admission, or for recurrent hematomas following the failure of a previous surgical evacuation. Decisions to proceed with embolization were made collaboratively by a multidisciplinary team, which included neurologists, neurosurgeons, intensive care specialists, internal medicine, geriatrics, and interventional neuroradiologists, following each center’s established protocols.

Anticoagulant and antiplatelet therapies were managed individually according to clinical criteria and institutional practices. In most cases, treatment was paused during the periprocedural period and reintroduced 24–72 h after embolization or surgery, depending on the hematoma’s stability and thromboembolic risk. As described in this study, embolization was often performed to facilitate the reintroduction of early anticoagulation when needed.

Corticosteroid therapy was not part of the embolization protocol in either center. However, a tapering regimen of dexamethasone was frequently administered to patients with CSDH during hospitalization, particularly those undergoing surgical evacuation. This practice varied depending on the admitting department and was not applied in a standardized or protocolized manner; therefore, it was not systematically recorded in the studied dataset.

Before embolization, all patients underwent a detailed clinical evaluation, and pre-procedural CT scans were reviewed to assess the characteristics of their CSDH. Hematoma thickness was measured using 0.625 mm coronal slice reconstructions, and other radiological parameters such as midline shift, sulcal effacement, and cerebral atrophy were documented.

The procedures were performed under general anesthesia, sedation, or local anesthesia. Procedural details, including the embolic agent used and technical specifications, were determined by the neuro interventionalist based on institutional protocols. Any procedural complications were also documented.

Both radial and femoral access routes were employed. Radial access was achieved under ultrasound guidance with a 5 under 6F short sheath, and it was accompanied by administering 4000 units of heparin and 2.5 mg of verapamil. Femoral access involved the use of a 6F sheath. A Select 5F Simmons 2 curve catheter (Penumbra, Alameda, CA, USA) was used for radial access, and an Envoy 6F (Cerenovus, New Brunswick, NJ, USA) for femoral access. Under biplane angiographic guidance, the common carotid artery (CCA) and external carotid artery (ECA) were selectively catheterized, with contrast injection from the CCA performed to evaluate the ophthalmic artery’s origin from the internal carotid artery (ICA). Selective contrast injection from the ICA was conducted in cases of uncertainty. The distal ECA was subsequently catheterized to allow for microcatheterization of the MMA using a Marathon (Medtronic, Toledo Way, Irvine, CA, USA) or a Headway duo (Terumo Neuro, Alton Pkwy, Irvine, CA, USA) microcatheter.

A super selective contrast injection was performed in the proximal MMA to map the artery and identify potentially dangerous connections. The microcatheter was advanced distally within the MMA, and 2 mg of lidocaine was injected in some cases to induce analgesia and vasodilation. Initially, Squid 12 was injected as the primary embolic agent. However, if the neuro interventionalist wanted to consider other techniques, different embolic materials could be used based on the specific case. These materials included ethylene vinyl alcohol copolymer (EVOH), *N*-butyl-2-cyanoacrylate (NBCA), and coils.

Follow-up CT scans were performed at 1, 3, and 6 months post-procedure to evaluate hematoma thickness, midline shift, sulcal effacement, and the need for surgical rescue. Treatment failure was defined as the need for rescue surgery, a hematoma thickness ≥ 10 mm on the 6-month follow-up scan, midline shift > 3 mm on the 6-month CT scan, or death related to the procedure. Readmissions were only recorded when directly related to the procedure. Other causes of readmission, such as unrelated medical issues, were not included in our dataset.

Statistical analysis was performed using IBM SPSS Statistics version 25 (IBM Corp, Armonk, NY, USA). In the case of quantitative variables, the median and interquartile range were calculated. For categorical variables, frequencies and percentages were calculated. We created two cohorts, one of all the patients included in this study and the other of all the hematomas treated. We divided them into patients who underwent surgery or not and hematomas that were treated surgically or not. To compare categorical variables between groups, we used the χ2 test and Fisher exact test, and when comparing continuous variables the Mann–Whitney U test was employed. For the comparison of quantitative variables such as the hematoma thickness at baseline and the follow-ups at 1, 3 and 6 months, the nonparametric Wilcoxon test was used for paired samples. In this way, each hematoma was compared with its own follow-up. The criterion established for considering a difference to be significant was a *p* ≤ 0.05.

## 3. Results

A total of 101 patients were treated with MMA embolization. Of these, 52 patients (51.5%) received a combination of embolization and surgery, while 49 (48.5%) were treated with embolization alone. The cohort included 69 men (68.3%) and 32 women (31.7%), with a median age of 82 years (IQR 75–87). No significant differences were observed between groups regarding their demographic or clinical variables (Table 1).

Radiological findings showed that patients in the combined-treatment group presented with a significantly larger initial hematoma thickness [initial right-side width (*p* = 0.031), initial left-side width (*p* = 0.002)] and larger midline shift (*p* = 0.002) than those treated with embolization alone. Additionally, patients in the combined-treatment group were those who exhibited symptoms more frequently (*p* = 0.000) (Table 1).

Concerning procedural variables, most patients (73, 72.3%) underwent the procedure under general anesthesia, which was the preferred method to ensure patient immobility during the injection of the embolic agent. Radial access was used in 54 patients (53.5%), while femoral access was employed in 47 patients (46.5%). In total, 42 of the hematomas (41.6%) treated were located on the left side, and 33 procedures (32.7%) involved bilateral embolization. Intra-arterial anesthesia was administered in 44 patients (43.6%) before embolization, with a median procedure duration of 53 min (IQR 42–77.5) (Table 2).

Six patients (5.9%) experienced complications, including one (1%) puncture site complication, two (2%) catheterization-related issues, and three (3%) complications associated with the embolization itself (Table 2). Four patients (4%) experienced treatment failure and were equally divided between the combined-treatment and embolization-only groups. Three cases of failure were attributed to a hematoma thickness exceeding 10 mm on the 6-month follow-up CT scan, while one patient treated solely with embolization required rescue surgery.

A total of 14 patients (13.9%) were lost to follow-up; 8 (7.9%) died before the 6-month follow-up CT scan, with 7 of these deaths occurring in the embolization-only group (*p* = 0.028). This higher mortality may be related to the increased complexity of these patients, who had higher comorbidity indices, and the fact that many of the procedures were carried out during the COVID-19 pandemic in Spain.

On the other hand, 134 hematomas (in the 101 patients previously reported) were analyzed. Of these, 61 (45.5%) were treated using a combination of embolization and surgery, and 73 (54.5%) were treated with embolization alone. Distal embolization was performed in 129 cases (93.6%). EVOH (Squid, Balt) was the preferred embolic agent used in 126 patients (94%) due to its favorable safety and efficacy profile (Table 3).

The reduction in hematoma thickness was analyzed for all patients within each treatment group at follow-up intervals of 1, 3, and 6 months. The median hematoma thickness at the 6-month follow-up was 0 across all groups (Figure 1). A significant reduction in hematoma size was noted over successive follow-up periods (*p* = 0.000). Moreover, the initial hematoma thickness was significantly greater in the combined-treatment group in this cohort as well, with a median of 18 mm (IQR 12–22.5) (*p* = 0.000). No statistically significant differences in hematoma thickness were found between the treatment groups in follow-up CT scans at 1, 3, and 6 months.

## 4. Discussion

This study evaluates the effectiveness of MMA embolization with Squid for managing CSDH, assessing its use as both a standalone procedure and an adjunct to surgery. In recent years, MMA embolization has gained prominence as a promising strategy in CSDH treatment, not only for relieving symptoms but also for addressing the underlying inflammatory mechanisms contributing to the persistence of hematomas. Early investigations by Ban et al. [10] highlighted that this approach significantly decreased treatment failure rates and reduced the need for repeat surgeries compared to traditional management. Similarly, Link et al. [7] reported favorable outcomes, while Kan et al. [12] demonstrated substantial hematoma reductions in over 90% of patients in a multicenter study, with minimal need for additional intervention. Salem et al. [13] further identified a smaller MMA diameter (<1.5 mm) to be a predictor of treatment failure, underscoring the artery’s role in the disease.

In line with these findings, our study demonstrates significant hematoma resolutions, with a median thickness of 0 mm at six months and no reinterventions required. These results reinforce the efficacy of MMA embolization as a minimally invasive alternative, particularly for patients who are poor candidates for surgery. Moreover, our findings highlight its adaptability, showing comparable outcomes when performed alone or alongside surgical evacuation, making it a versatile option for managing CSDH in diverse clinical scenarios.

Further research has explored the combined use of MMA embolization with surgery, especially in patients with recurrent or high-risk hematomas. Shotar et al. [9] demonstrated that combining embolization with surgery significantly reduced recurrence rates in patients prone to recurrence. Similarly, Ng et al. [14], Capatano et al. [15], and Onyinzo et al. [2] suggested that adjunctive embolization may offer additional benefits. Recently, Liebert et al. [11] evaluated MMA embolization’s effect on recurrent hematomas, finding that while the volume reduction was slower, it was ultimately more substantial on follow-up scans. In line with these findings, our study examines MMA embolization as a supplementary measure to enhance the durability of surgical decompression. Patients in our combined-treatment group presented with greater initial hematoma thickness and midline shift, showing the need for rapid volume reduction in more severe cases. Conversely, MMA embolization as a standalone therapy may be more appropriate for patients with mild symptoms who can tolerate a slower volume reduction. Despite these initial differences, the follow-ups showed comparable radiographic outcomes between groups, underscoring the importance of individualized patient selection based on clinical profiles and symptom severity.

Although MMA embolization has been shown to be effective, its limitations and the considerations for patient selection warrant discussion. Khorasanizadeh et al. [16] reported that combined treatment may involve higher initial costs, extended hospital stays, and increased complications. However, these outcomes were balanced by reduced recurrence rates and lower readmissions, which particularly benefit vulnerable populations. Catapano et al. [15] also noted cost savings due to fewer reinterventions. In our study, most patients tolerated MMA embolization well, with a low complication rate of 5.9%, consistent with the broader literature. However, a higher mortality rate was observed in the embolization-only group. Importantly, none of the deaths were attributable to the embolization procedure itself.

Recent randomized trials—EMBOLISE, STEM, and MAGIC—have provided further insights into the utility of MMA embolization. These studies are particularly relevant to our research due to their focus on liquid embolic agents. The EMBOLISE trial, led by Davies et al. [17], evaluated MMA embolization using Onyx (Medtronic) as the sole embolic agent. This trial demonstrated that combining MMA embolization with surgery significantly reduced the rate of hematoma recurrence or progression requiring reoperation (4.1% in the embolization group vs. 11.3% in the control group). Importantly, no embolization-related complications were reported, and the overall complication rates were comparable between groups. Similarly to our findings, mortality was higher in the embolization group.

The MAGIC trial, led by Liu et al. [18], examined MMA embolization with *Onyx* as well and identified specific subgroups that may have benefitted most, including patients managed without surgery, those with a midline shift less than 10 mm, and non-smokers. Our findings echo these observations, as the combined-treatment group presented with more severe symptoms and radiological findings, and standalone embolization may be effective in patients with less severe presentations. Additionally, MAGIC reported a lower incidence of serious adverse events with embolization, consistent with the safety profile observed in our cohort.

The STEM trial, conducted by Fiorella et al. [19], employed Squid (Balt) as the embolic agent, the same primary agent employed in our study. This trial reported a significant reduction in treatment failure rates with embolization compared to the control group (16% vs. 36%; *p* = 0.001), with the most notable benefits observed in patients managed non-surgically. These results underscore the importance of patient selection in optimizing success. Mortality was slightly higher in the embolization group (8% vs. 5%), but no significant differences were observed in functional outcomes at 180 days.

More trials on MMA embolization for CSDH are currently underway and are expected to provide further insights into the technique’s efficacy and safety. These studies may help refine patient selection criteria and optimize treatment protocols, contributing to establishing MMA embolization as a standard treatment option, especially for patients with a high recurrence risk or those unsuitable for surgery.

Our study has certain limitations. As a non-randomized prospective registry, it is subject to selection bias, which limits our ability to establish causal relationships. Differences in clinical protocols between the two participating centers, as well as among departments within each hospital, may have affected the consistency of both procedural approaches and follow-up practices. In particular, the lack of consensus regarding corticosteroid administration may have introduced variability; this has been acknowledged in the Materials and Methods section for clarity. Additionally, while the follow-up period was sufficient for evaluating primary outcomes, it may not have been long enough to detect late recurrences or complications. Patient loss at follow-up could also impact the reliability of recurrence assessments; most of the losses where due to the COVID-19 pandemic. Lastly, the relatively small sample size and diversity in patient profiles may limit the generalizability of our findings.

## 5. Conclusions

Our study adds to the growing body of evidence supporting MMA embolization with Squid as an effective and safe technique for managing CSDH, either as a standalone or adjunctive treatment. The findings indicate that surgery alone does not reduce hematoma thickness more than embolization alone, suggesting that embolization may be sufficient in selected cases. This is particularly relevant for patients whose hematoma thickness does not necessitate urgent surgical intervention, offering a less invasive option with potentially lower risks.

## Figures and Tables

**Figure 1 diagnostics-15-01424-f001:**
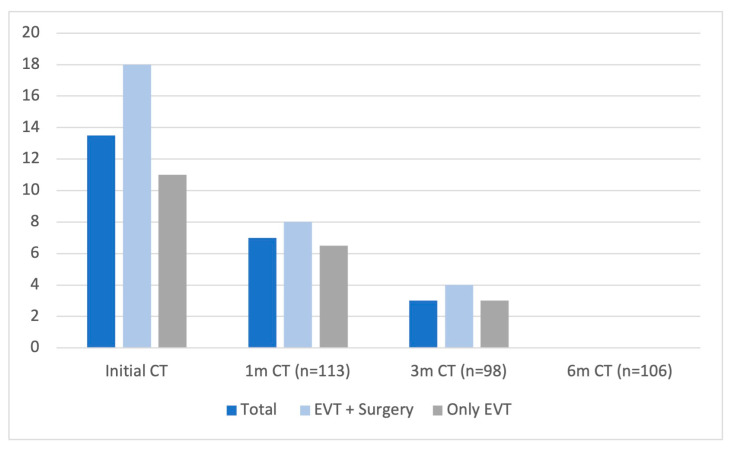
Reduction in hematoma thickness over time. Hematoma thickness is compared across three groups: total cases, the combined-treatment group, and the endovascular-only group. Measurements made at baseline (initial CT) and at 1-, 3-, and 6-month follow-up CT scans are presented. A progressive reduction in hematoma thickness is observed in all groups, with a median thickness of 0 mm at 6 months.

**Table 1 diagnostics-15-01424-t001:** Patients’demographic, clinical, and imaging variables.

	Total n = 101	EVT + Surgery n = 52 (51.5%)	Only EVT n = 49 (48.5%)	*p* Value
Men	69 (68.3%)	37 (71.2%)	32 (65.3%)	0.528
Women	32 (31.7%)	15 (28.8%)	17 (34.7%)	
Age	82 (75–87)	82 (75–87)	82 (74.5–87.5)	0.854
Smoker	32 (31.7%)	19 (36.5%)	13 (26.5%)	0.280
DM	35 (34.7%)	18 (34.6%)	17 (34.7%)	0.993
HTN	71 (70.3%)	35 (67.3%)	36 (73.5%)	0.498
DL	54 (53.5%)	29 (55.8%)	25 (51%)	0.633
Alcoholism	7 (6.9%)	5 (9.6%)	2 (4.1%)	0.438
CKD	13 (12.9%)	7 (13.5%)	6 (12.2%)	0.855
HF	24 (23.8%)	12 (23.1%)	12 (24.5%)	0.868
Stroke history	16 (15.8%)	6 (11.1%)	10 (20.4%)	0.222
Thrombocytopenia	3 (3%)	0 (0%)	3 (6.1%)	0.111
Coagulopathy	1 (1%)	0 (0%)	1 (2%)	0.485
APT	22 (21.8%)	14 (26.9%)	8 (16.3%)	0.197
AT	42 (41.6%)	17 (32.7%)	25 (51%)	0.062
Initial right-sidewidth (mm)	14 (10–18)	16 (10–24)	12.5 (10–14)	0.031
Initial left-side width (mm)	13 (10–19)	17 (12–22)	11 (8.25–16)	0.002
Symptomatic	81 (80.2%)	50 (96.2%)	31 (63.3%)	0.000
Right side	27 (26.7%)	14 (26.9%)	13 (26.5%)	0.979
Left side	42 (41.6%)	22 (42.3%)	20 (40.8%)	
Bilateral	32 (31.7%)	16 (30.8%)	16 (32.7%)	
Recurrence	8 (7.9%)	6 (11.5%)	2 (4.1%)	0.271
Cerebral atrophy	57 (56.4%)	29 (55.8%)	28 (57.1%)	0.889
MLS initial CT	49 (48.5%)	31 (59.6%)	18 (36.7%)	0.002
Sulcal effacement initial CT	67 (66.3%)	37 (71.2%)	30 (61.2%)	0.291

EVT: Endovascular Treatment; DM: Diabetes Mellitus; HTN: Hypertension; DL: Dyslipidemia; CKD: Chronic Kidney Disease; HF: Heart Failure; APT: antiplatelet therapy; AT: anticoagulant therapy.

**Table 2 diagnostics-15-01424-t002:** Patients’procedural variables and complications.

	Total n = 101	EVT + Surgery n = 52 (51.5%)	Only EVT n = 49 (48.5%)	*p* Value
Local anesthesia	2 (2%)	0 (0%)	2 (4.1%)	0.324
Conscious sedation	26 (25.7%)	13 (25%)	13 (26.5%)	
General anesthesia	73 (72.3%)	39 (75%)	34 (69.4%)	
Radial access	54 (53.5%)	24 (46.2%)	30 (61.2%)	0.129
Femoral access	47 (46.5%)	28 (53.8%)	19 (38.8%)	
Right-side EVT	26 (25.7%)	14 (26.9%)	13 (26.5%)	0.617
Left-side EVT	42 (41.6%)	20 (38.5%)	23 846.9%)	
Bilateral EVT	33 (32.7%)	18 (34.6%)	13 (26.5%)	0.668
I.A. anesthesia	44 (43.6%)	24 (46.2%)	20 (40.8%)	0.589
EVT duration (min)	53 (42–77.5)	53 (45–77)	51 (38.5–77.5)	0.306
Complications (total)	6 (5.9%)	3 (5.8%)	3 (6.1%)	1.000
Puncture site complications ^a^	1 (1%)	1 (1.9%)	0 (0%)	1.000
Catheterization complications ^b^	2 (2%)	1 (1.9%) *	1 (2%) **	1.000
Embolization complications ^c^	3 (3%)	1 (1.9%) ^†^	2 (4.1%) ^‡,¶^	0.610

EVT: Endovascular Treatment; ^a^ Right femoral artery pseudoaneurysm. ^b,^* Rupture of the frontal branch of the MCA during catheterization, with contrast extravasation into the diploic veins. Squid was injected. ** Thrombosis of the distal inferior division of the right M2 segment, with good collateral circulation. A bolus of Tirofiban was administered, followed by 2.5 mg of intra-arterial fibrinolysis in the affected branch, resulting in minimal residual thrombus in an M4 branch of the parietotemporal region. ^c,†^ VI cranial nerve palsy with spontaneous resolution within 24 h. ^‡^ Microcatheter rupture during removal from the parietal branch of the MMA following embolization. The proximal end of the microcatheter remained in the subclavian artery. The patient received 300 mg of aspirin and was placed on antiplatelet therapy for 30 days. ^¶^ Migration of Squid into a cortical vein during embolization of the parietal branch, observed on post-procedure CT imaging.

**Table 3 diagnostics-15-01424-t003:** Hematomas: procedure variables.

	Total n = 134	EVT + Surgery n = 61	Only EVT n = 73	*p* Values
Proximal EVT	5 (3.7%)	3 (4.9%)	2 (2.7%)	0.659
Distal EVT	129 (96.3%)	58 (95.1%)	71 (97.3%)	
Squid	126 (94%)	58 (95.1%)	68 (93.2%)	0.727
Squid 12	120 (89.6%)	52 (85.1%)	68 (93.2%)	0.136
Squid 18	11 (8.2%)	6 (9.8%)	5 (6.8%)	0.531
Squid + glue	1 (0.7%)	0 (0%)	1 (1.4%)	1.000
Squid + coils	4 (3%)	0 (0%)	4 (5.5%)	0.125
Glue	3 (2.2%)	3 (4.9%)	0 (0%)	0.092
Squid employed (cc)	0.4 (0.1–0.6)	0.3 (0.1–0.6)	0.4 (0.2–0.6)	0.087

EVT: Endovascular Treatment.

## Data Availability

Data is unavailable due to privacy or ethical restrictions.

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
