# Peer review of "Imaging and Clinical Outcomes Six Months After Middle Meningeal Artery Embolization with Squid for Chronic Subdural Hematoma: A Prospective Study"

_diagnostics, 2025, doi:10.3390/diagnostics15111424_

Round 1
Reviewer 1 Report
Comments and Suggestions for Authors
You have explained your chronic subdural etiopathogenesis well, but did you give steroid treatment to the patients in the preoperative and postoperative periods for inflammatory pathology? Please state in the method section. If you did or did not give steroid treatment, please state your views on this subject in the discussion section.
Secondly, both groups that received combined treatment and those that received only embolization consisted of elderly patients, and there were patients who used anticoagulant treatment in both groups. How did you ensure the anticoagulant treatment regimen during follow-up? Please specify your medical treatment scheme during follow-up
Author Response
Comments 1:
You have explained your chronic subdural etiopathogenesis well, but did you give steroid treatment to the patients in the preoperative and postoperative periods for inflammatory pathology? Please state in the method section. If you did or did not give steroid treatment, please state your views on this subject in the discussion section.
Response 1:
Thank you for your valuable observation. We are aware that, prior to the recent advances involving MMA embolization, the adjunctive use of corticosteroids or statins in patients undergoing surgical evacuation of chronic subdural hematomas (CSDH) had shown promising results, including reduced recurrence rates when compared with surgery alone. These pharmacologic strategies have been discussed in the literature as potential modulators of the underlying inflammatory process involved in CSDH pathophysiology.1–3
However, our study was not designed to evaluate the role of medical therapies such as corticosteroids in conjunction with MMA embolization. Consequently, we did not include corticosteroid therapy as part of our embolization treatment protocol, nor did we systematically collect data regarding their use. Therefore, corticosteroids were neither administered intentionally as part of the embolization strategy, nor were they a planned component of the study design.
That said, in our institution—and more broadly across the centers included in the study—patients with CSDH, particularly those undergoing surgical evacuation, often receive a tapering regimen of dexamethasone prescribed during hospitalization. This practice varies depending on the department or service managing the patient, and although common, it was neither standardized nor consistently documented across cases.
As corticosteroid use lies outside the scope and objectives of our study and given that it was neither a controlled variable nor included in our dataset, it was not analyzed. Nonetheless, we have now added a brief clarification in both the Methods (lines 99-104) and Discussion (lines 295-300) sections to reflect this practice, in line with the reviewer’s suggestion.
- Qiu S, Zhuo W, Sun C, et al. Effects of atorvastatin on chronic subdural hematoma: A systematic review. Medicine 2017; 96: e7290.
- Jiang R, Zhao S, Wang R, et al. Safety and Efficacy of Atorvastatin for Chronic Subdural Hematoma in Chinese Patients: A Randomized ClinicalTrial. JAMA Neurol 2018; 75: 1338.
- Yun H, Ding Y. How to remove those bloody collections: Nonsurgical treatment options for chronic subdural hematoma. Brain Circ 2020; 6: 254.
Comments 2:
Secondly, both groups that received combined treatment and those that received only embolization consisted of elderly patients, and there were patients who used anticoagulant treatment in both groups. How did you ensure the anticoagulant treatment regimen during follow-up? Please specify your medical treatment scheme during follow-up.
Response 2
Thank you for this pertinent observation. As described in the manuscript, one of the goals of embolization in selected patients was to enable the early reintroduction of anticoagulant therapy, particularly in elderly individuals with high thromboembolic risk.
Anticoagulation management was individualized based on clinical judgment, hematoma stability, and patient comorbidities. Although the specific treatment regimens were not systematically recorded, our analysis focused primarily on whether there were significant differences in outcomes—specifically recurrence—between patients who were anticoagulated or on antiplatelet therapy and those who were not. As reported in the Results section, no significant differences in recurrence rates were found between these groups.
We have now added a brief clarification in the Methods section to reflect the general approach to anticoagulation management during follow-up.
Reviewer 2 Report
Comments and Suggestions for Authors
We appreciate the authors in reporting their outcomes with MMA embolization with Squid for chronic subdural hematomas. The authors retrospectively compare embolization alone versus embolization with surgery. There is selection bias in these two groups with 96% of the MMA+surgery group symptomatic while only 68% or the MMA group. 59.6% of the surgery group had midline shift (ie mass effect) while only 36.7% of the embolization group. There have been RTC making these comparisons, EMBOLISE, STEM and MAGIC-MT trials. To add to the literature, it would be beneficial if they present additional information such as clinical outcomes (eg Modified Rankin Scale), other complications (seizures, ect). Otherwise, these findings are difficult to apply to the general population.
Author Response
Coments 1:
We appreciate the authors in reporting their outcomes with MMA embolization with Squid for chronic subdural hematomas. The authors retrospectively compare embolization alone versus embolization with surgery. There is selection bias in these two groups with 96% of the MMA+surgery group symptomatic while only 68% or the MMA group. 59.6% of the surgery group had midline shift (ie mass effect) while only 36.7% of the embolization group. There have been RTC making these comparisons, EMBOLISE, STEM and MAGIC-MT trials. To add to the literature, it would be beneficial if they present additional information such as clinical outcomes (eg Modified Rankin Scale), other complications (seizures, ect). Otherwise, these findings are difficult to apply to the general population.
Response 1.
We sincerely thank the reviewer for this thoughtful and constructive comment. We acknowledge the presence of selection bias between the treatment groups, which is indeed inherent to the design of this prospective observational study. As described in the manuscript, patients undergoing surgery in addition to embolization were those with more pronounced symptoms and radiological mass effect, such as midline shift. For this reason, treatment decisions were made on an individual basis by multidisciplinary teams, following clinical and imaging criteria. This is further discussed in the Discussion, where we explicitly highlight the existence of selection bias.
Regarding the suggestion to include functional outcome measures such as the modified Rankin Scale (mRS) or complication data (e.g., seizures), we appreciate its relevance. However, this study was conducted from an interventional neuroradiology perspective, focusing on procedural and radiological outcomes such as recurrence, hematoma evolution, and embolization safety, which fall within our area of follow-up and expertise.
In addition, these patients were admitted under the care of various departments (e.g., neurology, neurosurgery, geriatrics), each with potentially different standards for clinical evaluation. For this reason, functional outcomes such as mRS were not uniformly assessed and were not included in the analysis to maintain consistency and avoid introducing further variability.
While we are aware of the recent randomized trials (EMBOLISE, STEM, MAGIC-MT), we believe our work contributes meaningfully to the existing literature by providing prospective, real-world data on a relatively large cohort (101 patients) treated with Squid, a specific liquid embolic agent. As this technique continues to expand, we consider it important to reinforce and complement the available evidence.
Round 2
Reviewer 2 Report
Comments and Suggestions for Authors
We appreciate the authors making revisions to their study. Some questions still need to be addressed.
In abstract, authors state that there are clinical outcomes, yet their response is to report "procedural and radiographic outcomes".
In methods, although data is derived from different admitting groups, re-admission rates in both groupls should be reported in the list of complications.
In discussion, authors report that "...follow-up showed comparable outcomes between groups...". This should be clarified as comparable radiographic outcomes.
In discussion, authors report higher morbidity rates in embolization only group, and they attribute this to "higher baseline co-morbidities and external factors", yet the clinical demographic chart does not reflect this and shows comparable comorbidities. This conclusion is in valid.
Author Response
We sincerely thank the reviewer for the detailed and constructive comments. We have carefully revised the manuscript to address the points raised, as outlined below:
- “In abstract, authors state that there are clinical outcomes, yet their response is to report ‘procedural and radiographic outcomes’.”
Thank you for this observation. We have revised the abstract accordingly to ensure consistency and accuracy. The word “clinical” has been removed, and we now explicitly refer to “procedural and radiological outcomes”, in alignment with the scope of our study.
- “In methods, although data is derived from different admitting groups, re-admission rates in both groups should be reported in the list of complications.”
We thank the reviewer for this important suggestion. In our study, the only readmission considered was related to procedural failure, specifically one case requiring rescue surgery after embolization failure, as noted in line 191: “One patient treated solely with embolization required rescue surgery.”
Other causes of readmission, such as non-neurosurgical complications (e.g., pneumonia, urinary tract infections)—which are relatively common in elderly populations—were not included in our registry, as they fall outside the scope of the interventional outcomes we aimed to evaluate. Therefore, the only readmission recorded was related to neurosurgical reintervention.
We have clarified this point in the Methods section for transparency.
- “In discussion, authors report that '...follow-up showed comparable outcomes between groups...'. This should be clarified as comparable radiographic outcomes.”
We thank the reviewer for the clarification. We have amended the sentence in the Discussion to specify that the comparable outcomes refer to radiographic findings.
- “In discussion, authors report higher morbidity rates in embolization-only group, and they attribute this to ‘higher baseline co-morbidities and external factors’, yet the clinical demographic chart does not reflect this and shows comparable comorbidities. This conclusion is invalid.”
We appreciate this accurate and helpful observation. Although our clinical impression is that patients in the embolization-only group often presented with greater frailty and comorbidity, we recognize that this is not supported by statistically significant differences in our data.
Therefore, we have removed this statement from the Discussion to avoid drawing unsupported conclusions. We thank the reviewer for highlighting this point and helping us improve the clarity and accuracy of our interpretation.